# Air Quality Impacts of Smoke from Hazard Reduction Burns and Domestic Wood Heating in Western Sydney

**Maximilien Desservettaz** [1] , **Frances Phillips** [1] , **Travis Naylor** [1], **Owen Price** [2] , **Stephanie Samson** [2], **John Kirkwood** [3] **and Clare Paton-Walsh** [1,*]

1 Centre for Atmospheric Chemistry, University of Wollongong, Wollongong, NSW 2522, Australia; mjd232@uowmail.edu.au (M.D.); francesp@uow.edu.au (F.P.); naylort@uow.edu.au (T.N.)

2 Centre for Environmental Risk Management of Bushfire, University of Wollongong, Wollongong, NSW 2522, Australia; oprice@uow.edu.au (O.P.); ssamson@uow.edu.au (S.S.)

3 Office of Environment and Heritage, Lidcombe, NSW 2141, Australia; John.Kirkwood@environment.nsw.gov.au

\* Correspondence: clarem@uow.edu.au

**Abstract:** Air quality was measured in Auburn, a western suburb of Sydney, Australia, for approximately eighteen months during 2016 and 2017. A long open-path infrared spectrometer sampled path-averaged concentrations of several gaseous species, while other pollutants such as $PM_{2.5}$ and $PM_{10}$ were sampled by a mobile air quality station. The measurement site was impacted by a number of indoor wood-heating smoke events during cold winter nights as well as some major smoke events from hazard reduction burning in the spring of 2017. In this paper we compare the atmospheric composition during these different smoke pollution events and assess the relative overall impact on air quality from domestic wood-heaters and prescribed forest fires during the campaign. No significant differences in the composition of smoke from these two sources were identified in this study. Despite the hazard reduction burning events causing worse peak pollution levels, we find that the overall exposure to air toxins was greater from domestic wood-heaters due to their higher frequency and total duration. Our results suggest that policy-makers should place a greater focus on reducing wood-smoke pollution in Sydney and on communicating the issue to the public.

**Keywords:** long open-path FTIR; smoke; air quality

## 1. Introduction

Bushfires are a well-known natural hazard in Australia. Despite the focus of the media, it has been estimated that the death toll from air quality impacts may exceed those killed directly in the fires [1]. It is predicted that wildfire incidence will increase with higher temperatures as a consequence of climate change [2] which may increase the need for and the political will to undertake substantial hazard reduction burning, despite the significant negative impacts on air quality of prescribed fires [1,3–6]. The consequences of inhaled wood-smoke on human health (predominantly on the cardio-vascular system) are well documented in the literature (e.g., References [7–11]). Numerous studies have been undertaken to characterize the chemical composition of smoke from Australian fires ([12–29]) and the impacts of these fires on air quality in Australian cities (e.g., References [4,5,10,30]). There have been relatively few studies looking at the composition of wood-smoke pollution from domestic wood heaters in Australia and focus has been predominantly on rural communities, where this source dominates above all others [31–33]. Recent analysis has shown that in Sydney, domestic wood heaters are one of the most significant human contributors to wintertime fine particulate matter concentrations [34].

In this paper we describe measurements made as part of the Western Air-Shed Particulate Study for Sydney campaign at Auburn (WASPSS-Auburn). This campaign was driven by public interest and was designed to simulate a typical suburban balcony site in order to compare the pollution levels to those measured at nearby regulatory monitoring sites [35]. The focus of this paper is on the episodes of wood-smoke that were captured during WASPSS-Auburn, both from hazard reduction burning surrounding the Sydney basin and from domestic wood-heater pollution. Previous papers, from the WASPSS-Auburn campaign have described how the measurements shed light on the ammonia emissions from vehicles in the region [36] and compared the air quality at Auburn to the nearby regulatory air quality monitoring sites [35]. These papers describe the campaign in detail. WASPSS also included a roadside air quality study [37] and finalization of observations of previous campaigns [38–42] for use in a collaborative air quality modelling study (and intercomparison) of the greater metropolitan region of New South Wales [43–49].

During the Australian winter months, substantial increases in CO levels, coincident with increases in particulate matter and other trace gases, indicative of wood smoke, were noted during the cooler evenings at the WASPSS-Auburn measurement site. These occurred predominantly during June and July, the cooler months of the Australian winter and coincided with cold still nights. Wood burning stoves are a popular form of heating in Australian southern cities and we suggest these wood heaters to be the most likely source of these pollution events.

During the following two months (August and September), the air quality of Sydney was impacted on several occasions by smoke pollution from hazard reduction burns in the nearby forests. Sydney is bounded to the north, west and south by National Parks. Prescribed hazard reduction burns are lit during the cooler months in order to reduce the fuel load and limit the occurrence of wildfires during the summer months.

In this paper we analyse the measurements of atmospheric composition during the campaign with the aim of addressing two separate questions:

1. How comparable is the chemical composition of smoke from domestic wood-heaters to that from hazard reduction burns?
2. During the WASPSS-Auburn winter and spring of 2017, which of these sources of wood-smoke produced the greatest exposure to enhanced pollution levels in Auburn?

Our motivation to ask these questions arises from a hypothesis that the domestic wood-heater source provides a much more significant risk than is recognized by the public, unlike the highly publicized pollution events associated with hazard reduction burns.

The negative impact that exposure to toxic species such as CO and $PM_{2.5}$ can have on human health depends not only on the levels of toxins the person is exposed to but also the duration of the exposure. This study compares the relative exposure to pollutants in smoke from domestic wood heating and hazard reduction burns, by comparing the averaged enhancement of the toxin in the smoke and duration of the smoke events.

## 2. Methods

### 2.1. The Campaign

The WASPSS-Auburn campaign measurement site (see Figure 1) was established on the roof a 2 story building, on the edge of Auburn in Western Sydney and operated between 25 May 2016 and 15 September 2017. The site is adjacent to a major rail line, used for heavy diesel freight and local commuter trains and major road networks. To the east is a light industrial area, to the north and west is the Auburn central business district, with residential areas to the west. The greater Sydney metropolitan region is surrounded on all but one side by large forested regions: the Blue Mountains national park to the west (~35 km from the site), the Marramarra and Ku-rin-gai Chase national parks

to the north (∼20 km from the site) and the Royal and Heathcote national parks to the south (∼20 km from the site). The Pacific Ocean is also about 20 km from the site, to the east.

The site included a Mobile Air Quality (MAQ) station, comprising instrumentation measuring trace gases (CO, NO, $NO_2$, NOx, $O_3$ and $SO_2$), aerosols ($PM_{10}$ and $PM_{2.5}$), optical properties (Nephalometer) and meteorology (Wind speed and direction, temperature and relative humidity) (see Reference [35]). Also located adjacent to the MAQ station was an extended open-path Fourier transform infrared (OP-FTIR) spectrometer (see Reference [36]).The open-path instrument operated with a measurement path of 396 m, with the mirror arrays, which terminated the measurement path, located on the roof of a 3 story building, on a small hill, within the Auburn central business district (see Figure 1). The OP-FTIR spectrometer operated from October 2016 to March 2017 and May 2017 to September 2017 and, for the WASSPS-Auburn campaign, routinely targeted atmospheric $CO_2$, CO, $CH_4$ and $NH_3$. During periods of smoke pollution, levels of several trace gases of interest to urban air quality, are enhanced over the typical urban background levels and $CH_3OH$, $C_2H_2$, $C_2H_4$, $CH_2O$, indicative to wood smoke, were also retrieved from the FTIR spectra. In August 2017, an in-situ FTIR tracer gas analyser (CO, $CO_2$, $N_2O$, $CH_4$ and $_{13}C$ in $CO_2$) was installed with an air intake adjacent to the portable air monitoring station intake and operated until September 2017. Meteorological data were supplied by a 3D sonic anemometer from July 2017 to September 2017, which complimented the weather station of the portable monitoring station.

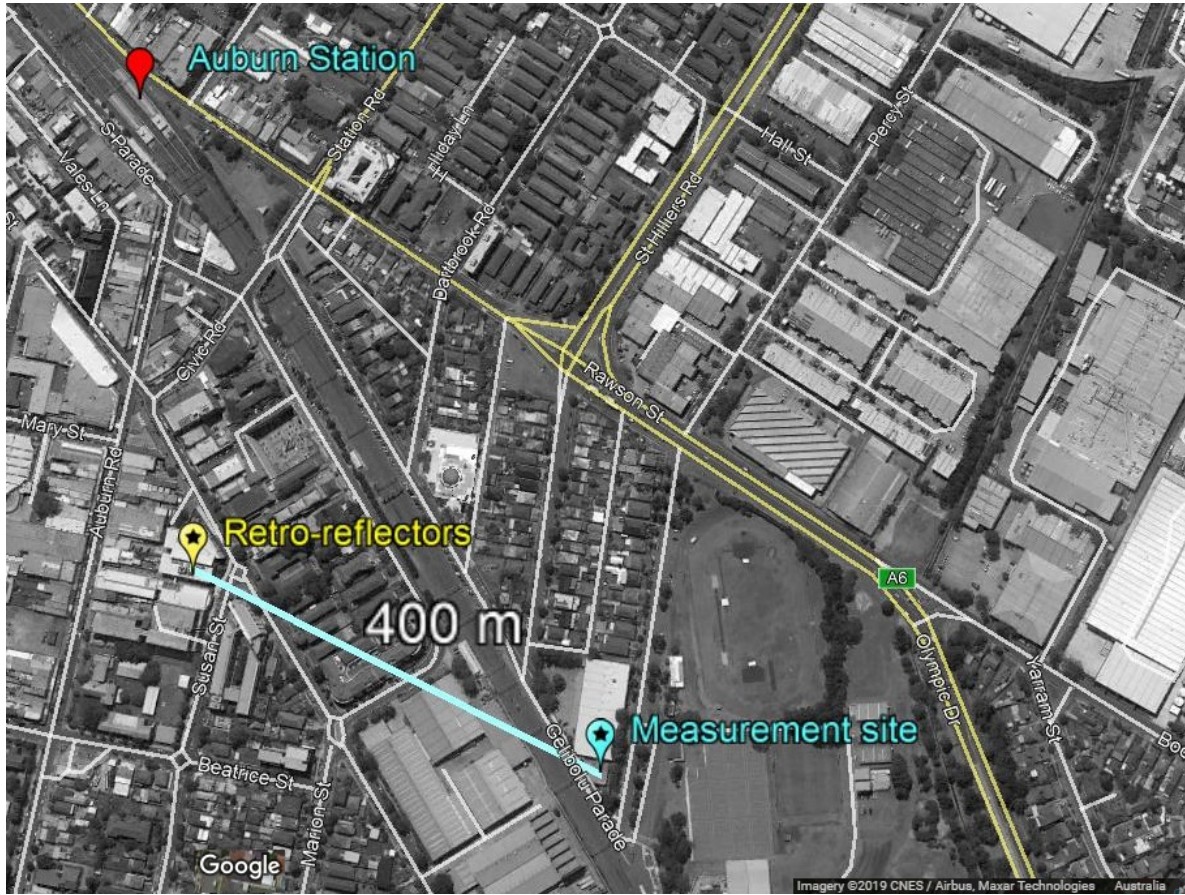

**Figure 1.** Locations over map (Source: Google Earth) of the Auburn measurement site, the retro-reflectors for the open-paths (400 m) and the Auburn train station for geographic reference.

## 2.2. Instrumentation

### 2.2.1. The Long Open-Path FTIR Spectrometer

The long open-path FTIR spectrometer (Matrix IR-Cube, Bruker Optik GmbH, Ettlingen, Germany) scans continuously recording a time-averaged (5-min) infrared absorption spectrum (500 to 4000 $cm^{-1}$) of the open atmospheric path, between the spectrometer and the retro-reflectors. The FTIR spectrometer is coupled to a 250 mm Schmidt-Cassegrain telescope (LX 200ACF, Meade Instruments Corporation, Irvine, CA, USA), modified to function as a parallel beam expander. The retro-reflectors consist of an array of 90 gold-plated corner cubes, which direct reflected light in the same direction as incoming radiation. The reflected radiation goes back through the telescope, refocusing into a narrow beam into the spectrometer's mechanically cooled (-196 C, RicorK508) MCT detector (Infrared Associates Inc., Stuart, FL, USA, or Judson Industries, Montgomeryville, PA, USA).

The infrared spectra, measured by the instrument, are analysed in micro-windows of the spectrum using MALT [50,51], a software program which uses quantum mechanical line strength and shape calculations from the HITRAN (or similar) database [52] to simulate reference spectra for each of the species with absorption lines within the micro-window, along with parameters of temperature, pressure and path length. MALT iteratively simulates spectra with different gas concentrations and line shape parameters to best match the measured spectrum. For this work, MALT returned path-averaged mixing ratios of $CO_2$, CO, $CH_3OH$, $NH_3$, $C_2H_2$, $C_2H_4$, $C_2H_6$, $CH_2O$ and water vapour [50]. The micro-windows used for the retrieval of the target species are detailed in Table 1. Figure 2 shows examples of a spectral fit for $CH_2O$, $CH_3OH$, $C_2H_4$ and $C_2H_2$,. Example fits for CO and $NH_3$ can be found in Phillips et al. [36]. The OP-FTIR system used here is described in further details, including an estimate of errors and validation, in the companion paper by Phillips et al. [36].

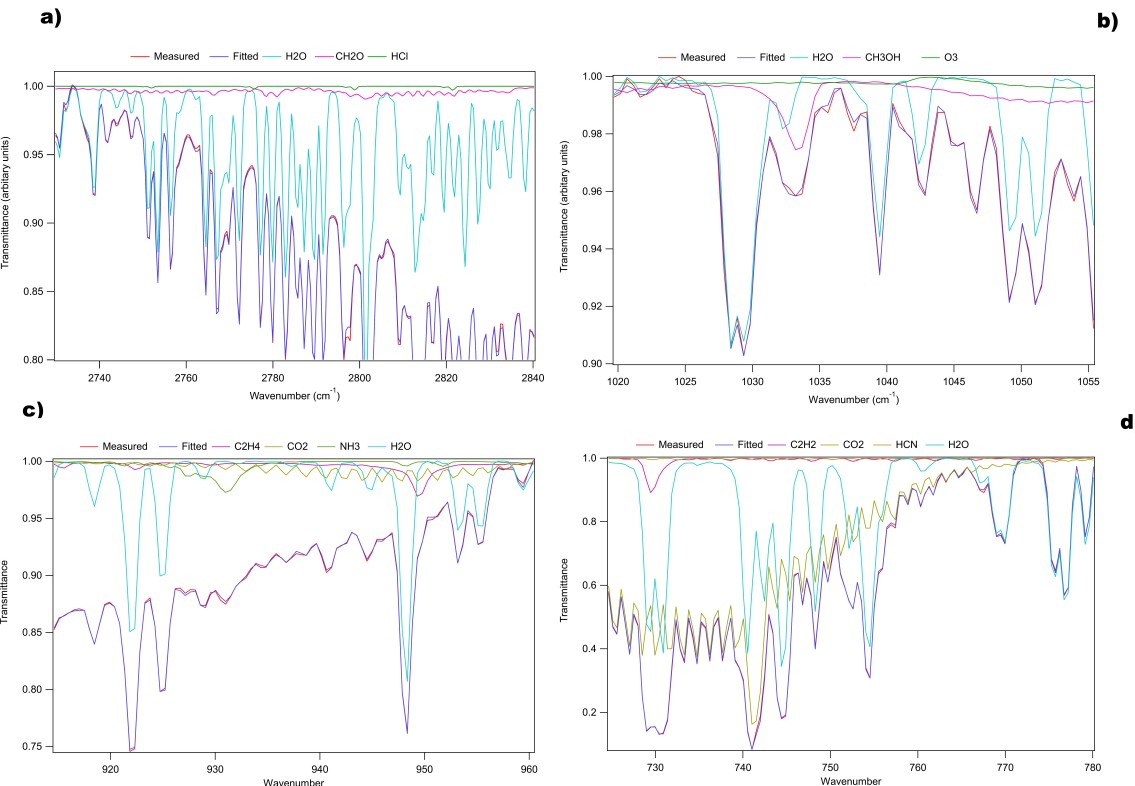

**Figure 2.** Example Multiple Atmospheric Layer Transmission (MALT) spectra fittings for (**a**) $CH_2O$, (**b**) $CH_3OH$, (**c**) $C_2H_4$ and (**d**) $C_2H_2$.

**Table 1.** Spectral Micro-windows limits used for the Multiple Atmospheric Layer Transmission (MALT) analysis of the Open Path-Fourier Transform Infrared (OP-FTIR) spectra.

| Target Gas Species | Micro-Window Wavelength Limits (cm$^{-1}$) | Interfering Species |
|---|---|---|
| CO, CO$_2$ | 2150–2280 | H$_2$O, N$_2$O |
| CH$_3$OH | 2010–1060 | NH$_3$, O$_3$, H$_2$O |
| * NH$_3$ | 900–945, 955–995 | H$_2$O |
| C$_2$H$_2$ | 710–760 | HCN, H$_2$O |
| C$_2$H$_4$ | 3001–3140 | H$_2$O |
| CH$_2$O | 2730–2840 | CH$_4$, HCl, H$_2$O |

* Analysis for NH3 was divided between 2 micro-windows to avoid a spectral artefact.

### 2.2.2. The Mobile Air Quality Station (MAQS)

The Mobile Air Quality Station (MAQS) is a mobile compact air quality station operated by the New South Wales Office of Environment and Heritage. The MAQS complies with the Australian standards for the measurement of ambient air quality. These standards are in accordance with the National Environmental Protection (Ambient Air Quality) Measure (NEPM). The MAQS is fitted with the following instruments:

- NO, NO$_2$, NOx and O$_3$ analyser (Teledyne, T204)
- SO$_2$ analyser (Teledyne, 100E)
- PM$_{2.5}$ and PM$_{10}$ analyser (Thermo Scientific, TEOM Series, 1405-DF)
- Temperature and humidity sensor (Vaisala, HMP155)
- Meteorology station (Met-One 50.5) for wind speed and wind direction.

Samples were taken at frequencies of 1-min and later averaged to 5-min to match the open-path FTIR's data frequency. Further details about the deployment, validation and data retrieved from the MAQS are presented in the companion paper by Simmons et al. [35].

### 2.3. Calculating Enhancement Ratios

As trace gases and particles emitted in biomass burning smoke, from domestic wood heating or bush fires, are co-emitted, there is a direct correlation between the emitted species. The nature of the fuel and the efficiency of the combustion control the relative amounts of the species emitted. This is reflected in the emission ratio, *ER*, where the concentration of a species, *i*, is referenced to a co-emitted species, *ref*:

$$ER_{i/ref} = \frac{\Delta[i]}{\Delta[ref]} = \frac{[i] - [i]_{Bgd}}{[ref] - [ref]_{Bgd}} \tag{1}$$

where [*i*] is the concentration of species *i*, in the presence of smoke and [*i*]$_{Bgd}$ is the "background" concentration, in the absence of wood smoke. Similarly [*ref*] and [*ref*]$_{Bgd}$ are the concentration of the reference species in the presence of wood smoke and in the background atmosphere; $\Delta_{[i]}$ and $\Delta_{[ref]}$ are the enhancements in *i* and *ref* species due to the smoke pollution. Typically the reference species is CO or CO$_2$, as the majority of carbonaceous material in the fuel is emitted either as CO or CO$_2$ [21]. CO is used as the reference species in this work, as the urban background of CO$_2$ was noted to be high and variable, possibly related to vehicle CO$_2$ emissions, resulting in greater uncertainty in the estimated background. The emission ratio gives a relative measure of the species emitted from burning biomass and allows emissions to be compared from different fuel sources and fire intensities. When smoke is sampled some distance from the emission point, this ratio is usually called an enhancement ratio, in recognition of the fact that chemical and physical processes cause quite rapid changes in the composition after emission.

An emission ratio or enhancement ratio can be retrieved from the slope of the linear regression between the species of interest, *i* and the reference species, *ref* [53]. This is useful when the background levels are difficult to define, as can be the case in an urban environment. A high coefficient of determination ($R^2$) represents a strong correlation between species *i* and the reference species, *ref* and suggests that influence from other emission sources is minimal [21,54].

The strength of the correlation between species *i* and the reference species, *ref*, can be weakened, following the emission event, due to extended residency within the boundary layer, with transport from the site of the fire or due to the influences of other emission sources [54]. In addition, ageing of the smoke (chemical removal or production) is likely to occur during transport of the smoke to the site. Therefore, retrieved ratios are not likely to represent pure emissions and the term enhancement ratio is used in the remainder of this study [54].

In this work, enhancement ratios referenced to CO measured at the Auburn site have been calculated for the species $CH_3OH$, $NH_3$, $C_2H_2$, $C_2H_4$ and $CH_2O$ measured by the OP-FTIR and NO, $NO_2$, NOx, $PM_{2.5}$, $PM_{10}$ and $SO_2$ measured by the MAQ station. While $CH_4$ and $C_2H_6$ are readily retrieved from the OP-FTIR infrared spectrum and are produced during wood burning, $CH_4$ and $C_2H_6$ are not included in this analysis due to considerable interference from other sources, believed to be dominated by leaks in the natural gas reticulation system in the urban environment.

## 3. Results

### 3.1. Smoke Events from Hazard Reduction Burns

Five major pollution events were recorded at the Auburn site between 1 August and 15 September 2017, characterised by rapid increases in CO and coincident increases in $PM_{2.5}$. Cross-referencing with the Rural Fire Services and media articles confirmed that these five pollution events were a result of hazard reduction burning in the forests surrounding Sydney. Figure 3, a satellite image of the greater Sydney region (source: https://worldview.earthdata.nasa.gov, date: 14 August 2017), shows the impact on Sydney of the smoke plume from the hazard reduction burn, sampled in our study from 13–15 August 2017. Adjacent is a picture featured in a news article reporting the impact of smoke, from this hazard reduction burn, on air quality in Sydney.

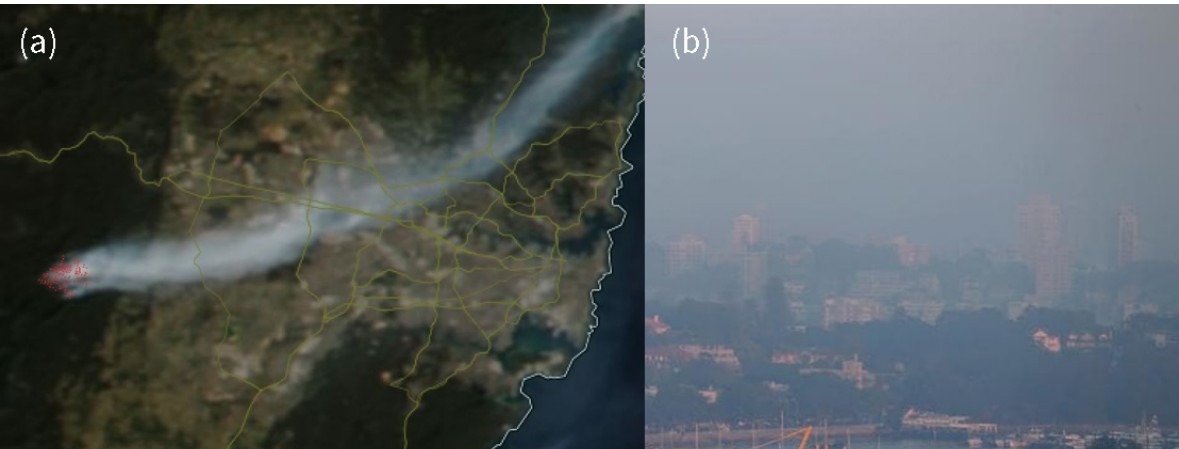

**Figure 3.** (**a**) MODIS image and thermal anomalies over Sydney on 14 August 2017 (Source: https://worldview.earthdata.nasa.gov) and (**b**) picture of smoke in Sydney from a news article citing the hazard reduction burns as the cause (Photo by John Grainger reproduced with kind permission from photographer. Source: News Corp Australia).

The five smoke events between August 1st and September 15th 2017 are attributed to specific hazard reduction burns, carried out by the NSW Rural Fire Services, together with burn area, distance and bearing from the Auburn site, in Table 2. The location of the hazard reduction burns, relative to the measurement site, are shown in Figure 4.

**Table 2.** Dates, name and number, burnt area and distance (bearing) from measurement site of each hazard reduction burn sampled at Auburn.

| Date | Fire Name (No) | Area (ha) | Distance (Bearing) from Site (km) |
|---|---|---|---|
| 13–15 August | HAW Ripple Creek HR (HR16070877281) | 2661 | 45 (W) |
| 20–23 August | HAW Burralow Road East HR (HR16090177857) | 409 | 49 (NW) |
| 26–27 August | HAW Campfire Creek HR (HR15120875086) | 467 | 40 (W) |
| 1–3 September | Moores Rd HR (HR14040968100) | 267 | 28 (N) |
| 9–13 September | Deep Bay HR (HR14042368385b) | 429 | 31 (NNE) |

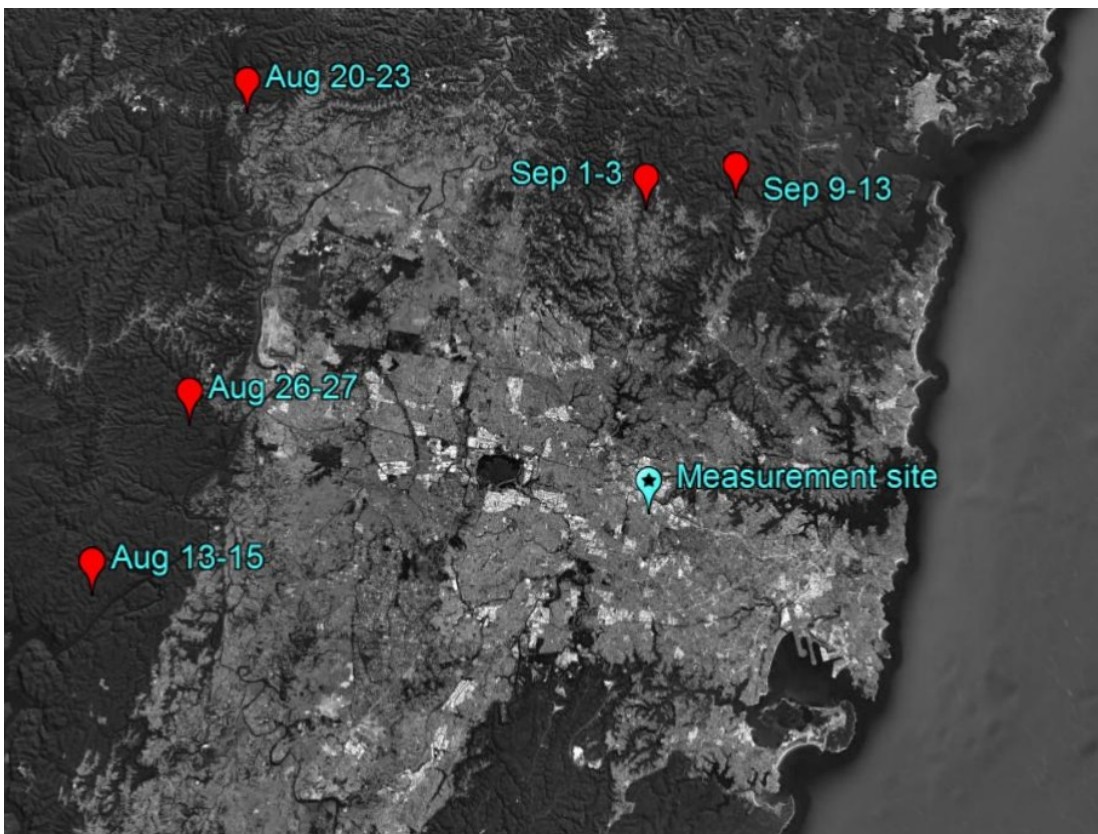

**Figure 4.** Location of hazard reduction burns and measurement site within the greater Sydney region (source: OEH; MODIS on https://worldview.earthdata.nasa.gov/. Google Earth.)

Figure 5 shows the CO and PM$_{2.5}$ time series between August and mid-September (when measurements finished), with the five hazard reduction burn smoke events shaded. In order to define when a smoke event has occurred, two factors were considered. An event was defined as occurring when peak CO exceeded 500 ppbv and was associated with strongly correlated coincident increases of PM$_{2.5}$. The threshold of 500 ppb CO is ten times larger than clean air background values in Sydney and this choice of threshold excludes some minor pollution peaks that are poorly correlated with increases in PM$_{2.5}$. The longest event (around September 11th) showed intermittent enhancements due to changing meteorology (possibly as a result of the common sea breezes experienced in this region) [42,46]. Instead of separating this event into smaller events, the event is only considered to have ended when both CO and PM$_{2.5}$ have returned to background levels for several hours. The duration of each of the hazard reduction burns events defined this way are shown by the shaded areas in Figure 5 and lasted between 17 and 80 h.

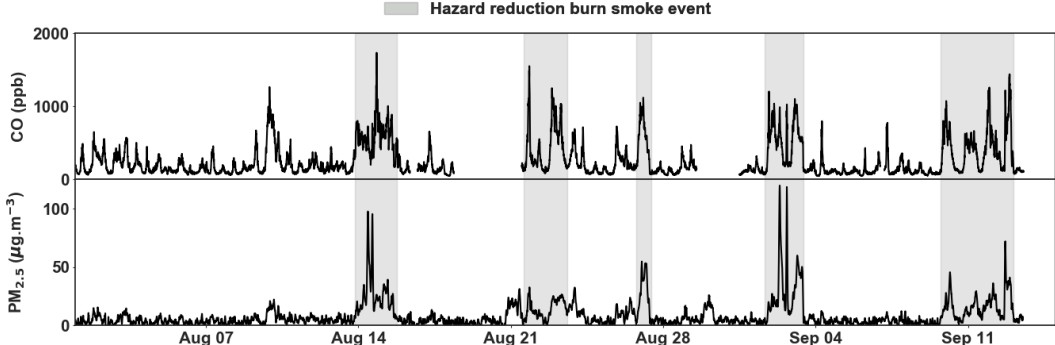

**Figure 5.** Time series of CO and PM$_{2.5}$ during August and first half of September 2017. Shading highlights hazard reduction burn smoke events.

### 3.2. Smoke Events from Domestic Wood Heating

In June and July 2017 many overnight pollution events impacted the Auburn measurement site. These events were characterised by high concentrations of CO and coincident increases of PM$_{2.5}$, much like the hazard reduction burn smoke events. All these events started in the late afternoon and were characterised by lower temperatures ($\leq 15\ ^\circ$C), promoting the use of heaters and low wind speeds ($\leq 2\ \text{m·s}^{-1}$), promoting accumulation in the nocturnal boundary layer. Figure 6 shows the time-series for CO and PM$_{2.5}$, evening temperature and wind speed (averaged between 16:00 and 22:00) during June and July 2017. Applying the same criteria that was used to identify hazard reduction burn smoke events (i.e., peak CO concentrations above 500 ppb and strongly correlated increases in PM$_{2.5}$), 42 events were identified that were assumed to be the result of domestic wood-heating pollution in the region.

The use of a relatively high threshold for CO helps to ensure that pollution events from other sources are not included but will most likely underestimate the pollution from domestic wood-heaters by excluding weaker events that are then included in the non-event background values. All these events began in the evening, between 16:00 and 18:00, however, the time at which trace gases and particulate matter concentrations returned to typical urban background levels, varied from as early as 22:00 the same evening to after 04:00 the following day. Due to the difficulty of defining an exact duration for each event and for consistency, all domestic wood heating smoke events were considered to last from 16:00 to 04:00. These events are shown by the shading in Figure 6. The fact that all the events occur overnight shows the importance of the reduced mixing layer depth during the cold nights when domestic wood-heaters are commonly used.

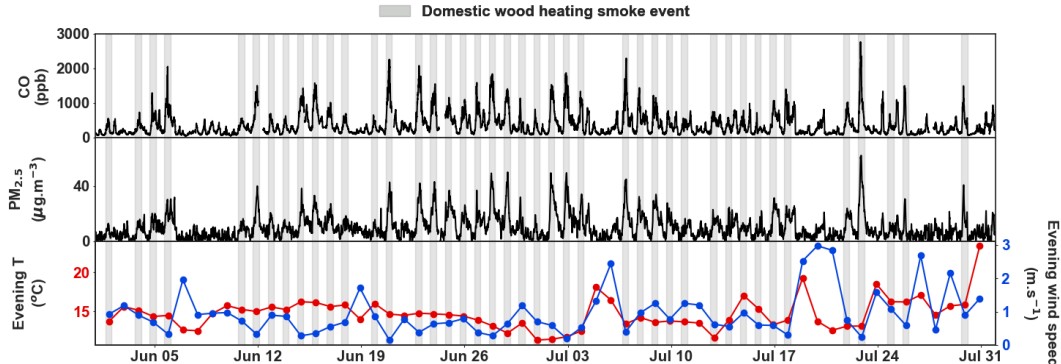

**Figure 6.** Time series of CO, PM$_{2.5}$, evening temperature and wind speed (averaged between 16:00 and 22:00) during June and July 2017. Shading highlights domestic wood heating smoke events.

### 3.3. Background Concentrations

In order to estimate the enhanced concentrations of atmospheric pollutants during smoke events, we first need to estimate an appropriate "background" concentration to subtract from the concentrations measured during an event. We considered a number of possible ways of estimating an appropriate background for each source of smoke, including:

1. average concentrations throughout the campaign excluding the times when there was an identified event;
2. average night-time (16:00–04:00) concentrations throughout the campaign excluding the times when there was an identified event;
3. average daytime (04:00–16:00) concentrations throughout the campaign excluding the times when there was an identified event; and
4. average night-time (16:00–04:00) concentrations during June and July excluding the nights when there was a domestic wood-heater smoke event

The different estimations of "background" concentrations of pollutants calculated these ways are given in a table in the appendix (Table A1). We initially assumed that the average night-time concentrations during the winter months of June and July (excluding the nights with domestic wood-heater events) would make the most suitable background for the domestic wood-heater events. However, we found that these estimates were likely biased low, since they used nights that were unusually windy. For this reason we assumed that the best estimate for "background" concentrations for both sources of smoke was the average concentration throughout the campaign excluding smoke events, which we refer to as (*Study no event*). The standard deviation of this background amount has been used to estimate the uncertainty in the calculated average enhancement as it dominates all other relevant uncertainties. It should be noted that there will also be some systematic uncertainties, which are discussed in References [21,36] but that these will be identical for both smoke sources and so do not impact the comparison or conclusions of the study.

### 3.4. Enhanced Concentrations of Pollutants during Smoke Events

Table 3 presents the average concentrations of pollutants throughout the campaign excluding the times when there was an identified event (*Study no event*); the average concentrations during the nights with domestic wood heating events (*DWH event*) and the average concentrations of pollutants during hazard reduction burn events (*HRB event*). Also shown are the resulting average enhancements in pollutant concentrations during domestic wood-heater (ΔDWH) and hazard reduction burn events (ΔHRB), calculated from the difference of the two. ΔDWH is the average over the 42 domestic wood heater smoke events and represents 426 h of measurement data. ΔHRB is the average of 5 identified smoke events due to hazard reduction burns, representing the average of 234 h of measurement data.

The enhancements, reported in Table 3, represent the average increased exposure, for the nearby population, over the duration of the smoke events. It is worth noting that, while maximum measured concentrations during HRB events were greater than those measured in the domestic wood burning smoke, the measured concentrations were more varied and occurred over a longer duration, compared to the domestic wood smoke events.

From Table 3, the average enhancement in levels of CO and NO are significantly higher during the domestic wood heater smoke events compared with the hazard reduction burn events. In contrast $PM_{2.5}$ and $PM_{10}$ are greater in the hazard reduction burn smoke. The higher levels of NO suggests that locally produced domestic wood heating smoke is fresher (less aged) than that of hazard reduction burns, where smoke has travelled further to reach the sampling site. The lower enhancements in $PM_{2.5}$ and $PM_{10}$ in the domestic heater smoke may be the result of filters on the wood heaters or better combustion efficiency.

**Table 3.** Averaged concentrations of pollutants throughout the campaign in the absence of smoke events and during domestic wood-heater and hazard reduction burn events. Averaged concentrations in the absence of smoke events are subtracted from concentrations during the events to give the respective averaged enhancements during domestic wood heating (ΔDWH) and hazard reduction burns (ΔHRB). Uncertainties quoted are standard deviations of the average concentrations. The standard deviations in the average background amounts are used to estimate the uncertainties in the enhancements as the choice of background to subtract is the dominant uncertainty.

| | Study | DWH | HRB | | |
| --- | --- | --- | --- | --- | --- |
| | No Event | Event | Event | ΔDWH | ΔHRB |
| CO (ppb) | $220 \pm 170$ | $640 \pm 420$ | $500 \pm 290$ | $420 \pm 170$ | $280 \pm 170$ |
| $CH_3OH$ (ppb) | $3 \pm 1$ | $5 \pm 2$ | $6 \pm 3$ | $2 \pm 1$ | $3 \pm 1$ |
| $NH_3$ (ppb) | $3 \pm 2$ | $7 \pm 4$ | $6 \pm 4$ | $4 \pm 2$ | $2 \pm 2$ |
| $C_2H_2$ (ppb) | $2 \pm 1$ | $3 \pm 2$ | $3 \pm 2$ | $1 \pm 1$ | $1 \pm 1$ |
| $C_2H_4$ (ppb) | $3 \pm 2$ | $7 \pm 4$ | $6 \pm 3$ | $4 \pm 2$ | $3 \pm 2$ |
| $CH_2O$ (ppb) | $4 \pm 1$ | $6 \pm 2$ | $6 \pm 2$ | $2 \pm 1$ | $2 \pm 1$ |
| NO | $16 \pm 30$ | $50 \pm 58$ | $37 \pm 43$ | $34 \pm 30$ | $22 \pm 30$ |
| $NO_2$ (ppb) | $15 \pm 12$ | $25 \pm 12$ | $27 \pm 13$ | $10 \pm 12$ | $12 \pm 12$ |
| NOx | $31 \pm 38$ | $77 \pm 68$ | $64 \pm 52$ | $46 \pm 38$ | $33 \pm 38$ |
| $PM_{2.5}$ ($\mu g/m^3$) | $6 \pm 6$ | $17 \pm 11$ | $22 \pm 17$ | $11 \pm 6$ | $15 \pm 6$ |
| $PM_{10}$ ($\mu g/m^3$) | $13 \pm 8$ | $26 \pm 14$ | $31 \pm 19$ | $13 \pm 8$ | $19 \pm 8$ |
| $SO_2$ (ppb) | $0.9 \pm 0.7$ | $2 \pm 1$ | $2 \pm 1$ | $0.7 \pm 0.7$ | $1.0 \pm 0.7$ |

### 3.5. Cumulative Enhancements

To compare the overall pollution burden from hazard reduction burning and domestic wood-heaters, we need to calculate the cumulative enhancements of different pollutants from each source. We define the cumulative enhancement as the average concentration of the pollutant during an event, $Av[i]_{Event}$, minus the average concentration of the pollution when there is no event, $Av[i]_{UrbanBgd}$, multiplied by the total duration of all the events (i.e., total number of hours of all events). We can define the cumulative enhancement of species $i$, $(CE_i)$ as:

$$(CE_i) = Av\Delta[i] \times time = Av([i]_{event} - [i]_{UrbanBgd}) \times time \tag{2}$$

where $Av(\Delta[i])$ is the average enhancement in pollutant $i$ and *time* is in hours.

It should be noted that the cumulative enhancement calculation is not very sensitive to the assumed duration of each event. This is because the enhanced concentration is averaged over the assumed duration of the event. This means that if an event actually lasted 6 h and then returned to background values and the assumed event was 12 h, then the cumulative enhancement (in ppb. hours) would be the same (since the assumed enhanced concentration would be half the true enhanced concentration and the assumed duration would be double the true duration). This makes the comparison of the cumulative enhancement a better measure than a simple comparison of the average enhanced concentrations in smoke as shown in Table 3.

The impact on health from exposure to air toxics, such as particulate matter, is aggravated when it is extended in time and repeated. As such the concept of cumulative enhancements best represents the added exposure to pollutants for the public in the areas impacted by the smoke pollution and we can use the terms "cumulative enhancements" and "cumulative exposure" interchangeably.

Table 4 presents the cumulative exposure to the species studied here from both domestic wood heating and hazard reduction burning, between 1 June and 15 September 2017. For instance, the cumulative added exposure to CO from domestic wood heating is $\Delta DWH_{CO}$ (420 ppb) $\times$ 426 h, which equals to 178,920 ppb·h. The table also presents the ratio between the two sources for each species, as a means of comparing the relative contribution to cumulative exposure from each source of smoke over this period.

**Table 4.** Cumulative exposure (enhancements) from domestic wood heating (DWH), hazard reduction burn (HRB) and the ratio between the two (DWH : HRB).

| Species | DWH (426 h) | DWH:HRB | HRB (234 h) |
|---|---|---|---|
| CO (ppb·h) | 178,920 | 2.7:1 | 65,520 |
| $CH_3OH$ (ppb·h) | 852 | 1.2:1 | 702 |
| $NH_3$ (ppb·h) | 1704 | 3.6:1 | 468 |
| $C_2H_2$ (ppb·h) | 426 | 1.8:1 | 234 |
| $C_2H_4$ (ppb·h) | 1704 | 2.4:1 | 702 |
| $CH_2OH$ (ppb·h) | 852 | 1.8:1 | 468 |
| NO (ppb·h) | 14,484 | 2.8:1 | 5148 |
| $NO_2$ (ppb·h) | 4260 | 1.5:1 | 2808 |
| NOx (ppb·h) | 19,596 | 2.5:1 | 7722 |
| $PM_{2.5}$ ($\mu g·h/m^3$) | 4686 | 1.3:1 | 3510 |
| $PM_{10}$ ($\mu g·h/m^3$) | 5538 | 1.2:1 | 4446 |
| $SO_2$ (ppb·h) | 298 | 1.3:1 | 234 |

*3.6. Calculating Enhancement Ratios to Compare the Chemical Composition of Smoke from Different Sources*

Using the linear regression method described in Section 2.3, enhancement ratios, referenced to CO, $ER_{i/CO}$, were calculated for the 42 identified domestic wood-heater events, for all species of interest. For each event, enhancement ratios were calculated for each species over four time intervals: 16:00 to 22:00, 16:00 to 00:00, 16:00 to 02:00 and 16:00 to 04:00. This was done in order to identify the longest sampling period with the strongest correlations between pollutants, because this is likely to represent the best observations of the smoke plume, uncontaminated by pollution from other sources. For each species, the duration with the highest $R^2$ value (within 10% of the maximum between the 4 periods) was noted. For each event, the duration that included the strongest correlations for the greatest number of species was selected for the analysis of all species. To avoid interference from other local sources, events where the $R^2$ value of $ER_{PM_{2.5}/CO}$, $ER_{PM_{10}/CO}$, $ER_{CH_3OH/CO}$ and $ER_{C_2H_4/CO}$ were less than 0.5, were removed from the analysis, to provide a dataset that best represented the composition of smoke from domestic wood heaters. This meant that our analysis of smoke composition used only the clearest (most highly correlated) 16 of the original 42 smoke events, to provide the final, reported, $ER_{i/CO}$.

Similarly, in calculating the enhancement ratios, for the hazard reduction burn smoke events, shorter periods of the longest smoke event were chosen to encompass the strongest enhancements, in order to achieve higher coefficients of determination ($R^2 > 0.6$), that is, the 80-h long smoke event was analysed over two shorter duration periods, which showed the strongest enhancements. The analysis was limited to periods where the correlations between trace gases and particulates to the reference gas was strongest to avoid contamination from other sources and thereby more accurately characterise and compare the chemical composition of smoke from the different fire sources.

Enhancement ratios for the targeted species in the smoke from domestic wood heating and hazard reduction burns, derived from the linear regression and referenced against CO are presented in Table 5, together with the averaged $R^2$ values from the linear regressions. This represents 16 of the 42 identified domestic wood burning smoke events and the 6 analysed periods for the 5 hazard reduction burn smoke events. For all species reported in this work, the differences in enhancement ratios for the smoke produced by the domestic wood heaters and the hazard reduction burns were not significant.

## 4. Discussion

*4.1. Chemical Composition*

From our measurements we find very similar chemical composition within the smoke from hazard reduction burns and from the domestic wood-heater pollution events. Enhancement ratios relative to the reference gas, CO, were calculated from the linear regression (see Table 5). The differences in the

chemical composition of the smoke from the two sources are not significant (differences are smaller than the combined standard deviation). This finding implies that the health impacts from the two types of pollution will likely be similar for a given pollution concentration and duration. It also provides evidence to confirm our original assumptions that the night-time pollution events result from smoke from wood burning. Since our study only covers a limited number of pollutants, it is still possible that there are differences in the chemical composition of the smoke with respect to other gases not detected by our measurements.

**Table 5.** Linear Regression enhancement ratios of trace gases and particulate matter against CO retrieved from domestic wood heating and hazard reduction burn smoke periods. Those enhancement ratios were retrieved from a selected 16 domestic wood heating and 5 hazard reduction burns events. The uncertainties quoted are the standard deviations of the average of all events. Whilst there will be other systematic uncertainties, these will be the same for both sources of smoke and therefore will not effect the comparison

| Target Species | Domestic Wood Heating $ER_{i/CO}$ | $R^2$ | Hazard Reduction Burns $ER_{i/CO}$ | $R^2$ | Australian Forest Fires [21,23,29,33] |
|---|---|---|---|---|---|
| $CH_3OH$ | $0.005 \pm 0.002$ | $0.9 \pm 0.1$ | $0.007 \pm 0.002$ | $0.8 \pm 0.1$ | $0.017 \pm 0.005$ |
| NH3 | $0.009 \pm 0.003$ | $0.9 \pm 0.1$ | $0.007 \pm 0.003$ | $0.7 \pm 0.2$ | $0.023 \pm 0.006$ |
| $C_2H_2$ | $0.004 \pm 0.001$ | $0.8 \pm 0.1$ | $0.005 \pm 0.003$ | $0.6 \pm 0.1$ | not reported |
| $C_2H_4$ | $0.009 \pm 0.001$ | $0.9 \pm 0.1$ | $0.008 \pm 0.001$ | $0.9 \pm 0.1$ | $0.016 \pm 0.008$ |
| $CH_2O$ | $0.004 \pm 0.002$ | $0.8 \pm 0.1$ | $0.006 \pm 0.003$ | $0.8 \pm 0.1$ | $0.023 \pm 0.007$ |
| NO | $0.10 \pm 0.03$ | $0.7 \pm 0.1$ | $0.14 \pm 0.02$ | $0.8 \pm 0.2$ | not reported |
| $NO_2$ | $0.04 \pm 0.02$ | $0.7 \pm 0.1$ | $0.04 \pm 0.01$ | $0.7 \pm 0.1$ | not reported |
| NOx | $0.12 \pm 0.04$ | $0.7 \pm 0.1$ | $0.13 \pm 0.05$ | $0.7 \pm 0.2$ | not reported |
| $PM_{2.5}$ | $0.02 \pm 0.01$ | $0.8 \pm 0.1$ | $0.03 \pm 0.01$ | $0.6 \pm 0.1$ | not reported |
| $PM_{10}$ | $0.03 \pm 0.01$ | $0.8 \pm 0.1$ | $0.04 \pm 0.01$ | $0.7 \pm 0.1$ | not reported |
| $SO_2$ | $0.002 \pm 0.001$ | $0.6 \pm 0.1$ | $0.002 \pm 0.001$ | $0.7 \pm 0.2$ | not reported |

The data presented in Table 5 uses only the times when all the pollutant concentrations are most highly correlated with CO, in order to minimise the possibility of the measurements being distorted by pollution from other sources. It is also possible to calculate enhancement ratios by ratioing the cumulative enhancements of each pollutant $\Delta_i$ as calculated by Equation (2) (and shown in Table 3) with the cumulative enhancement of CO, $\Delta_{CO}$. This data is shown in the appendix (Table A2). The resulting enhancement ratios are very similar for the domestic wood-heater events (except for lower $NO_2$) but for the hazard reduction burns the results show more $PM_{2.5}$, $PM_{10}$, $CH_3OH$ and $SO_2$ and less NO. This demonstrates the advantage of the linear regression method, which does not require an estimate of the background, which can change rapidly in an urban environment.

Comparing the enhancement ratios estimated here (using the preferred linear regression method) with emission ratios from previous studies of Australian forest fires smoke composition [21,23,29,33] (see Table 5), most enhancement ratios, $ER_{i/CO}$ from this study are between 2 and 3 times lower than emission ratios reported in the literature.

The enhancement ratios reported here are measured away from the source and over the duration of the smoke event, whilst the emission ratios from the literature in Table 5 were measured immediately following emission at the combustion source. The lower enhancement ratios in this study are most likely the result of chemical ageing (removal) between emission and sampling. Our results reinforce the conclusion that only emission ratios and emission factors measured very close to the fires should be considered in compilations of emission factors used for emissions modeling [54].

### 4.2. Immediate and Cumulative Exposure

Exposure to poor air quality can have immediate negative health impacts, such as increased hospital admissions during pollution events (e.g., References [1,10]). However there is also evidence

of increased morbidity as a result of cumulative exposure to air pollutants (e.g., References [9,11,55]). Therefore both immediate and cumulative exposure are important when considering the impacts of smoke on air quality in Sydney. The enhanced concentrations of CO and NO were substantially higher during the domestic wood heater events than during hazard reduction burn pollution events. However the greater impact on human health is from particulate matter [56], with concentrations of $PM_{2.5}$ and $PM_{10}$ in the hazard reduction smoke significantly greater than in the domestic wood heater smoke. The data here shows that hazard reduction burns lead to greater added enhancements in $PM_{2.5}$ and $PM_{10}$, having potentially a worse immediate effect on health. The lower amounts of particulate matter during wood-heater smoke events point to the potential importance of filtering mechanisms on wood heaters.

Another reason to be concerned about wood-smoke pollution is that a large number of pollutants are emitted together, with the potential for additive effects as the body is exposed to a toxic mixture of chemicals. A number of pollutants measured in this study have been identified as belonging to the same toxicological class, meaning that they attack the human body in the same way [57]. For instance CO, $C_2H_2$, $C_2H_4$ and $C_2H_6$ all disrupt oxygen transport in the body, whilst $SO_2$, $NO_2$, $CH_2O$ and $NH_3$ all cause both eye irritation and upper respiratory tract irritation [57]. There are a very large number of other pollutants that are known to be associated with smoke pollution [54] that were not measured as part of this study and these will also add to the health impacts of populations exposed to pollution from smoke events.

Cumulative exposure includes the total duration of exposure to smoke pollutants. During this study, 426 h of domestic wood smoke events were recorded, compared with 234 h of hazard reduction burn events. Taking into account the duration of exposure to the smoke pollutants, cumulative exposure from smoke from domestic wood burning heaters is consistently higher than smoke from hazard reduction fires.

The timing of smoke events should also be considered when comparing exposure to the population. Domestic wood heating smoke events occur from 16:00 until at least 22:00, affecting the population during the evening commuter peak times and outdoor evening activities, such as sports training. Hazard reduction smoke events occur throughout the day and are, therefore, likely to impact a greater fraction of the population. However, Australian residential buildings are reported to be permeable to air toxins and during the evening toxins could impact some fraction of Sydney's population [58].

Hazard reduction burns typically occur during the spring (March to May) and autumn (August to September) months. The OP-FTIR did not operate between March 18th and May 23rd and 10 weeks of the spring hazard reduction season are not included in this analysis. The mobile air quality station operated for the full spring period. However a study of smoke plumes captured by the MODIS instruments on-board cameras counted 45 plumes within 60 km of the Auburn measurement site during 2017, 36 of which took place during our sampling period, between 1 August and 15 September 2017. This suggests that, potentially, 80% of hazard reduction burns in 2017 took place within the sampling period. While the mapping of smoke plumes is limited by satellite retrieval, which rely on low frequency overpass (twice daily) and can be hindered by cloud coverage, this provides an estimate of the fraction of hazard reduction burn events sampled during the year.

In any case, the number of smoke events impacting Sydney will vary from year to year, dependent on the number and location of prescribed burns, the meteorology transporting and trapping the smoke over the city and the number of cold winter nights enticing the use of wood burning heaters. While the study here was of limited duration, it does provide a good comparison of the composition of smoke from domestic wood burning heaters and hazard reduction burns as well as an indication of the likely overall impact on the population of Sydney from these two pollution sources.

## 5. Summary and Conclusions

In this study we analysed atmospheric composition data from the WASPSS-Auburn campaign at times when the site was impacted by smoke pollution from hazard reduction burns and from domestic wood-heaters. We set out to use these data to answer the following questions:

1. How comparable is the chemical composition of smoke from domestic wood-heaters to that from hazard reduction burns?
2. During the WASPSS-Auburn winter and spring of 2017, which of these sources of wood-smoke produced the greatest exposure to enhanced pollution levels in Auburn?

No significant differences in the chemical composition of smoke from pollution events from wood-heaters and from hazard reduction events were found, from analysis using enhancement ratios from the smoke events with the most highly correlated increases in different pollutants. We found that the peak concentrations of particulate matter were higher during the hazard reduction burns, implying a greater immediate threat to the health of vulnerable members of the population. Nevertheless, the extended overall duration of the domestic wood-heater events meant that the cumulative exposure from these pollution events exceeded that of the hazard reduction burns during the study period. Whilst the relative pollution from these two sources of wood-smoke in Sydney will vary from year to year, our results highlight the significance of pollution from domestic wood-heaters in Sydney.

The problem of pollution from hazard reduction burns is well-known and the fire services are beginning to implement methods to reduce the likelihood of smoke from burn-offs impacting local populations. In future, more should be done to reduce pollution from wood-heaters in Sydney and to communicate the issue more broadly to the public and policy-makers.

**Author Contributions:** Conceptualization, C.P.-W.; methodology, F.P., T.N. and J.K.; formal analysis, F.P., M.D. and O.P.; investigation, M.D.; data curation, F.P. and J.K.; writing—original draft preparation, M.D., F.P. and C.P.-W.; writing—review and editing, all authors; visualization, M.D.; supervision, C.P.-W.; project administration, C.P.-W.; funding acquisition, C.P.-W.

**Funding:** This research was supported by the Australian Government's National Environmental Science Program through the Clean Air and Urban Landscapes Hub and from the NSW Office of Environment and Heritage via the Bushfire Risk Management Research Hub.

**Acknowledgments:** The Authors would like to acknowledge the NSW Master Plumbers Association and the Auburn and Cumberland City Council for providing the locations for the measurement sites and to Paul Naylor and Douglas Greening of the NSW Master Plumbers Association for their technical assistance in the operation of the instruments. The authors also wish to acknowledge the work of the staff of the NSW Office of Environment and Heritage and EPA, in particular Gunaratnam Gunashanhar, for the maintenance of the EPA Portable Monitoring Station, plus Doreena Dominick and Chris Caldow, from the Centre for Atmospheric Chemistry, for assistance with the maintenance of the spectrometers.

**Conflicts of Interest:** The authors declare no conflict of interest.

## Abbreviations

The following abbreviations are used in this manuscript:

| | |
|---|---|
| FTIR | Fourier Transform InfraRed |
| ERs | Emission/Enhancement Ratios |
| MAQS | Mobile Air Quality Station |
| DWH | Domestic Wood HEating |
| HRB | Hazard Reduction Burn |

# Appendix A

**Table A1.** Different background concentrations tested in calculation of enhancements.

| | Study No Event | Night Time (16:00–4:00) No Event | Day Time (4:00–16:00) No Event | DWH Period Night Time No Event |
|---|---|---|---|---|
| CO | $220 \pm 170$ | $230 \pm 170$ | $220 \pm 170$ | $190 \pm 90$ |
| $CH_3OH$ (ppb) | $3 \pm 1$ | $3 \pm 1$ | $3 \pm 1$ | $3 \pm 1$ |
| $NH_3$ (ppb) | $3 \pm 2$ | $4 \pm 2$ | $3 \pm 2$ | $3 \pm 2$ |
| $C_2H_2$ (ppb) | $2 \pm 1$ | $1 \pm 1$ | $2 \pm 1$ | $1 \pm 1$ |
| $C_2H_4$ (ppb) | $3 \pm 2$ | $3 \pm 2$ | $3 \pm 2$ | $2 \pm 1$ |
| $CH_2O$ (ppb) | $4 \pm 1$ | $5 \pm 1$ | $4 \pm 1$ | $4.4 \pm 0.9$ |
| NO (ppb) | $16 \pm 30$ | $12 \pm 29$ | $18 \pm 30$ | $8 \pm 22$ |
| $NO_2$ (ppb) | $15 \pm 12$ | $18 \pm 14$ | $13 \pm 10$ | $16 \pm 13$ |
| NOx (ppb) | $31 \pm 38$ | $30 \pm 38$ | $32 \pm 37$ | $24 \pm 32$ |
| $PM_{2.5}$ ($\mu g/m^3$) | $6 \pm 6$ | $7 \pm 6$ | $6 \pm 5$ | $5 \pm 4$ |
| $PM_{10}$ ($\mu g/m^3$) | $13 \pm 8$ | $12 \pm 7$ | $13 \pm 9$ | $11 \pm 5$ |
| $SO_2$ (ppb) | $0.9 \pm 0.7$ | $0.8 \pm 0.7$ | $0.9 \pm 0.7$ | $0.6 \pm 0.6$ |

**Table A2.** Enhancement ratios of trace gases and particulate matter against CO retrieved from domestic wood heating and hazard reduction burn smoke periods. Those enhancement ratios were retrieved from a selected 16 domestic wood heating and 5 hazard reduction burns events through the *Linear Regression* method, as well as from all 42 domestic wood heating and 5 hazard reduction burn events using the *Ratio of Averages* method.

| Target Species | Domestic Wood Heating | | | Hazard Reduction Burns | | |
|---|---|---|---|---|---|---|
| | $ER_{i/CO}$ (Linear Fit) | $R^2$ | $ER_{i/CO}$ ($\Delta i/\Delta CO$) | $ER_{i/CO}$ (Linear Fit) | $R^2$ | $ER_{i/CO}$ ($\Delta i/\Delta CO$) |
| $CH_3OH$ | $0.005 \pm 0.002$ | $0.9 \pm 0.1$ | $0.005 \pm 0.003$ | $0.007 \pm 0.002$ | $0.8 \pm 0.1$ | $0.011 \pm 0.008$ |
| $NH_3$ | $0.009 \pm 0.003$ | $0.9 \pm 0.1$ | $0.009 \pm 0.006$ | $0.007 \pm 0.003$ | $0.7 \pm 0.2$ | $0.008 \pm 0.009$ |
| $C_2H_2$ | $0.004 \pm 0.001$ | $0.8 \pm 0.1$ | $0.003 \pm 0.003$ | $0.005 \pm 0.003$ | $0.6 \pm 0.1$ | $0.004 \pm 0.005$ |
| $C_2H_4$ | $0.009 \pm 0.001$ | $0.9 \pm 0.1$ | $0.009 \pm 0.005$ | $0.008 \pm 0.001$ | $0.9 \pm 0.1$ | $0.010 \pm 0.009$ |
| $CH_2O$ | $0.004 \pm 0.002$ | $0.8 \pm 0.1$ | $0.004 \pm 0.003$ | $0.006 \pm 0.003$ | $0.8 \pm 0.1$ | $0.007 \pm 0.006$ |
| NO | $0.10 \pm 0.03$ | $0.7 \pm 0.1$ | $0.08 \pm 0.08$ | $0.14 \pm 0.02$ | $0.8 \pm 0.2$ | $0.1 \pm 0.1$ |
| $NO_2$ | $0.04 \pm 0.02$ | $0.7 \pm 0.1$ | $0.02 \pm 0.03$ | $0.04 \pm 0.01$ | $0.7 \pm 0.1$ | $0.04 \pm 0.05$ |
| NOx | $0.12 \pm 0.04$ | $0.7 \pm 0.1$ | $0.1 \pm 0.1$ | $0.13 \pm 0.05$ | $0.7 \pm 0.2$ | $0.1 \pm 0.2$ |
| $PM_{2.5}$ | $0.02 \pm 0.01$ | $0.8 \pm 0.1$ | $0.03 \pm 0.02$ | $0.03 \pm 0.01$ | $0.6 \pm 0.1$ | $0.06 \pm 0.04$ |
| $PM_{10}$ | $0.03 \pm 0.01$ | $0.8 \pm 0.1$ | $0.03 \pm 0.02$ | $0.04 \pm 0.01$ | $0.7 \pm 0.1$ | $0.07 \pm 0.05$ |
| $SO_2$ | $0.002 \pm 0.001$ | $0.6 \pm 0.1$ | $0.002 \pm 0.002$ | $0.002 \pm 0.001$ | $0.7 \pm 0.2$ | $0.004 \pm 0.003$ |

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
