# Peer review of "Air Quality Impacts of Smoke from Hazard Reduction Burns and Domestic Wood Heating in Western Sydney"

_atmosphere, doi:10.3390/atmos10090557_

Round 1

Reviewer 1 Report

Reviewer comments

No: atmosphere-576472

Title: Air quality impacts of smoke from hazard reduction burns and domestic wood heating in western Sydney

This paper is very interesting. However, because too many assumptions and implications have been used to draw conclusions, they should be rewritten more clearly. Therefore, this MS could not be published in Atmosphere.

Minor comments:

Please add full name on abbreviations when first used (ex: WASPSS). Although you have created a section for the abbreviations in the back, the full name on a number abbreviations were not included. Line 21-22: Please explain more detail effects on human health by these references. Take a deeper look at the implications of the results of the OP-FTIR. Some figures and tables are not explained in the text. (ex: Figure 2, Figure 6) Unify terms used in illustrations and explanations. (ex: H2CO vs CH2O) Line 147-181: Please add references. 174-175: Why did you use 500 ppbv of CO as standard. Provide justification for setting these criteria. There are many hypotheses and assumptions in the methods part. More relevant references and explanations are needed to support this. Many of the content in the methods and results is redundant. Minimize duplicates. The contents of the table title and body are too duplicated. Correct it more concisely. Line 307-308: In this paper, authors did not study many substances to compare the composition of pollutants. In particular, since the concentration of CO is very high compared to other pollutants, it is difficult to distinguish between events by the ER value of the target substances. Line 336-338: Provide references. The contents of Tables 5 and 6 are almost identical. Clean up again.

Author Response

We thanks all four reviewers for their time and helpful comments. We believe that we have addressed all the comments raised. We have rearranged the sections of the paper in a more logical fashion and hope that readers will now find the manuscript clearer and more concise. We have reproduced all the comments below in bold and given our response to each point in regular text below.

Reviewer 1

Title: Air quality impacts of smoke from hazard reduction burns and domestic wood heating in western Sydney

This paper is very interesting. However, because too many assumptions and implications have been used to draw conclusions, they should be rewritten more clearly. Therefore, this MS could not be published in Atmosphere.

Unfortunately this reviewer does not elucidate as to what assumptions are considered problematic and therefore it is not possible for us to respond to this non-specific criticism.

Minor comments:

Please add full name on abbreviations when first used (ex: WASPSS). Although you have created a section for the abbreviations in the back, the full name on a number abbreviations were not included.

Line 1 The reference to WASSPS has been removed from the abstract.

Line 29 The full name for WASPSS is included at the first instance use in the main manuscript.

Line 21-22: Please explain more detail effects on human health by these references.

We have not elaborated further on human health aspects because we consider it outside the focus of the paper (and the journal). Our study does not collect any data on human health impacts and the health impacts are mentioned simply in regards to the interest in air quality.

Take a deeper look at the implications of the results of the OP-FTIR.

It is unclear what the reviewer means by this. Other results from this novel set of measurements have been reported elsewhere (as described and referenced in this manuscript: see line 36 and reference 36).

Some figures and tables are not explained in the text. (ex: Figure 2, Figure 6)

Figure 2 is now referenced this way:

Line 108 MALT iterates through species concentrations and line shape parameters to best match the measured spectrum (see Figure 2) has been changed to

MALT iteratively simulates spectra with different gas concentrations and line shape parameters to best match the measured spectrum.

And at Line110 added:  Figure 2 shows examples of a spectral fit for CH2O, CH3OH C2H4 and C2H2. Example fits for CO and NH3 can be found in Philips et al (2019).

Caption for Figure 2 Example fits for CO and NH3 can be found in Philips et al (2019). Has been removed.

Figure 6 is now referenced this way:

Line 229 The following has been added: The location of the hazard reduction burns, relative to the measurement site, are shown in Figure 6.

Unify terms used in illustrations and explanations. (ex: H2CO vs CH2O)

Figure 2 has been updated to CH2O to match the remaining manuscript

Line 147-181: Please add references.

References added to Wooster et al, 2011; Paton-Walsh et al 2014 and Akagi et al , 2011.

174-175: Why did you use 500 ppbv of CO as standard. Provide justification for setting these criteria.

We have reordered the manuscript to present the hazard reduction burn events first. This allows for a clearer explanation of the choice of the 500 ppb CO threshold.

The relevant sentences now read:

Figure 5 shows the CO and PM$_{2.5}$ time series between August and mid-September (when measurements finished), with the five hazard reduction burn smoke events shaded.  In order to define when a smoke event has occurred, two factors were considered. An event was defined as occurring when peak CO exceeded 500 ppbv and was associated with strongly correlated coincident increases of PM$_{2.5}$. The threshold of 500 ppb CO is ten times larger than clean air background values in Sydney and this choice of threshold excludes some minor pollution peaks that are poorly correlated with increases in PM$_{2.5}$. The five hazard reduction burn events identified this way lasted between 17 and 80 hours.

There are many hypotheses and assumptions in the methods part. More relevant references and explanations are needed to support this.

I am not aware of another study that compared smoke enhancements in this way to reference, however the reviewer’s comment that there are many assumptions and hypotheses is over-stated.

The 3 assumptions made are these:

Threshold for identifying events (this has been explained in the text along with the consequence for bias – which would be to under-estimate the domestic wood-heater pollution) The duration of the events (i.e. time over which we average the concentrations). This is also explained in the text, along with an explanation of why the results are insensitive to the choice of event duration. The “background” amount that we choose to subtract to calculate the enhancement. This is the greatest assumption and carries that largest uncertainty through the results. Again, this is clearly explained in the text.

Further, we disagree with the reviewer’s emphasis on the importance of the assumptions made. The result that we arrive at is that the wood-smoke pollution (during this particular time-frame) is slightly larger than that for bushfire events. We do not make any claim that this would be the same for all time-frames or make any statement that over-reaches the relevance of our results. In the end we are only using the data to make a qualitative statement that wood-smoke is an important source of pollution in Sydney that has been widely overlooked by the public in discussions about the city’s air quality. There are no assumptions inherent in our methods that would change this basic conclusion.

Many of the content in the methods and results is redundant. Minimize duplicates.

The methods and results have been reorganised and simplified, including deleting Table 5. The reorganisation of the manuscript allows for a clearer flow of the ideas and redundant paragraphs have been removed.

The contents of the table title and body are too duplicated. Correct it more concisely.

The following changes have been made:

Line 108 remove Figure 2 reference; add at line 110 “\textbf{Figure \ref{Fig:Fittings} shows examples of a spectral fit for CH$_2$O, CH$_3$OH, C$_2$H$_2$ and C$_2$H$_4$.}” Line 188 to 196 :42 domestic wood heating smoke events were identified when CO levels exceeded 500 ppbv, in association with similar increase in PM$_{2.5}$. All events start in the late afternoon, between 16:00 and 18:00, however, the time at which trace gases and particulate matter return to urban background levels varies between 22:00 the same evening and 04:00 the following day.

Figure 3 shows the 5-min averaged time series for CO and PM$_{2.5}$, for June and July 2017, together with the meteorological variables of temperature and wind speed, averaged for the period between 16:00 and 22:00. The identified domestic wood heating smoke events, indicated by shading over the time series, are characterised by lower temperatures ($\leq$ 15$^{\circ}$C), promoting the use of heaters, and low wind speeds ($\leq$ 2 m s$^{-1}$), promoting accumulation in the boundary layer.

Has been Changed to:

42 domestic wood heating smoke events were identified when CO levels exceeded 500 ppbv, in association with similar increase in PM2.5 (indicated by shading over the CO and PM2.5  time series in Figure 3). All events start in the late afternoon, between 16:00 and 18:00, however, the time at which trace gases and particulate matter return to urban background levels varies between 22:00 the same evening and 04:00 the following day. The identified domestic wood heating smoke events,  are characterised by lower temperatures ($\leq$ 15$^{\circ}$C), promoting the use of heaters, and low wind speeds ($\leq$ 2 m s$^{-1}$), promoting accumulation in the boundary layer.

Caption Figure 3 now reads: “Time series of CO, PM2.5, evening temperature and wind speed (averaged between 16:00 and 22:00) during June and July 2017. Shading highlights domestic wood heating smoke events.”

Line 307-308: In this paper, authors did not study many substances to compare the composition of pollutants. In particular, since the concentration of CO is very high compared to other pollutants, it is difficult to distinguish between events by the ER value of the target substances.

The fact that the concentration of CO is high does not mean that differences cannot be seen in the emission ratios if they exist. It just changes the order of magnitude of the emission ratio.

It is true that there may be significant differences in substances not measured in this study. We have added the following sentence:

“Since our study only covers a limited number of pollutants, it is still possible that there are differences in the chemical composition of the smoke with respect to other gases not detected by our measurements.”

Line 336-338: Provide references.

Reference to Lelieveld,2015, Shah, 2013 and Caldwell 1988 added

Line 336 now reads: However the greater impact on human health is from particulate matter (Lelieveld, et al, 2015, Shah, 2013 and Caldwell 1988) with concentrations of PM2.5 and PM10 in the hazard reduction smoke significantly greater than in the domestic wood heater smoke.

The contents of Tables 5 and 6 are almost identical. Clean up again.

We have reorganised the manuscript, deleting Table 5. 

Reviewer 2 Report

The manuscript presents high quality research to assess an important environmental issue. I have only a few suggestions for the Authors' consideration before publishing the paper.   Please provide details on the calibration and the measurement error of the devices in Section 2.1. 

A Google logo is necessary in Fig. 1, as required by the Google Earth license.

Republishing Fig. 2b might require an explicit permission from News Corp Australia. Please check the license there.    I do not fully agree with the arbitrary selection of the event duration. As Authors correctly state, the cumulative enhancement calculation should not depend much on the duration of each event. Furthermore, near-background nighttime concentrations a few hours after an "event" can be reasonably regarded as remainders of the ageing plume, as mixing in a stable nighttime boundary layer is very weak. Ageing also decreases correlations, thus an "event" might get closed too early by the R2 method. Therefore, for better comparability and generality, I would recommend to use overnight enhancements in all cases.    Surprisingly, the mixing layer depth is never mentioned in the manuscript, although it is the most important meteorological parameter affecting nighttime pollution levels. I recommend to obtain the nighttime inversion depths from the nearest radiosonde site and plot it as a fourth graph in Fig.3. This information together with wind velocity could also help to find analogous nights for background definition, which was honestly and correctly identified by the Authors as the main bottleneck of this study.    In the Acknowledgements, please use capital A in "Centre for atmospheric Chemistry".    Overall, the manuscript is good and I believe this is already acceptable for publication in its present form. I leave in the Authors' consideration whether they further improve the paper according to my suggestions.   

Author Response

We thanks all four reviewers for their time and helpful comments. We believe that we have addressed all the comments raised. We have rearranged the sections of the paper in a more logical fashion and hope that readers will now find the manuscript clearer and more concise. We have reproduced all the comments below in bold and given our response to each point in regular text below.

Reviewer 2

Comments and Suggestions for Authors

The manuscript presents high quality research to assess an important environmental issue. I have only a few suggestions for the Authors' consideration before publishing the paper.   Please provide details on the calibration and the measurement error of the devices in Section 2.1. 

It is not possible to make a real “calibration” in open-path geometry. We have now made more explicit references to validation descriptions in Phillips et al, 2019 (line 113) and Simmons et al, 2019 (line 127)

A Google logo is necessary in Fig. 1, as required by the Google Earth license.

Thanks1 - Figure 1 has been modified to show the google logo as suggested.

Republishing Fig. 2b might require an explicit permission from News Corp Australia. Please check the license there. 

  Thanks – we already had permission but we have adjusted figure caption to make this more explicit

I do not fully agree with the arbitrary selection of the event duration. As Authors correctly state, the cumulative enhancement calculation should not depend much on the duration of each event. Furthermore, near-background nighttime concentrations a few hours after an "event" can be reasonably regarded as remainders of the ageing plume, as mixing in a stable nighttime boundary layer is very weak. Ageing also decreases correlations, thus an "event" might get closed too early by the R2 method. Therefore, for better comparability and generality, I would recommend to use overnight enhancements in all cases.

Overnight enhancements (16:00 – 04:00) were already used for all 42 domestic wood-heater events to estimate enhancements. The R2 test was only used to select the events with the strongest correlations for comparison of smoke composition. We have rearranged the manuscript substantially and clarify this point now explicitly.

Surprisingly, the mixing layer depth is never mentioned in the manuscript, although it is the most important meteorological parameter affecting nighttime pollution levels. I recommend to obtain the nighttime inversion depths from the nearest radiosonde site and plot it as a fourth graph in Fig.3.

This information together with wind velocity could also help to find analogous nights for background definition, which was honestly and correctly identified by the Authors as the main bottleneck of this study. 

While this is a reasonable request from Reviewer 2, and the inclusion of the boundary layer information would be beneficial to the manuscript, radio sonde data is only available from Sydney Airport, on Botany Bay, which is heavily influenced by the coastal meteorology. The boundary layer data from Sydney airport therefore would not represent conditions at our site.

 We have added another sentence to more explicitly mention the importance of reduced mixing in the nocturnal boundary layer:

This sentence “The identified domestic wood heating smoke events, are characterised by lower temperatures (<15C), promoting the use of heaters, and low wind speeds (< 2ms-1) promoting accumulation in the boundary layer. “

Has been replaced by these:

“The identified domestic wood heating smoke events, are characterised by lower temperatures (<15C), promoting the use of heaters, and low wind speeds (< 2ms-1) promoting accumulation in the nocturnal boundary layer. The fact that all the events occur overnight shows the importance of the reduced mixing layer depth during the cold nights when domestic wood-heaters are commonly used.“

 In the Acknowledgements, please use capital A in "Centre for atmospheric Chemistry". 

Corrected 

Overall, the manuscript is good and I believe this is already acceptable for publication in its present form. I leave in the Authors' consideration whether they further improve the paper according to my suggestions.    

Reviewer 3 Report

Abstract: "WASPSS" replace with "Western Air-Shed Particulate Study for Sidney". Misprint (page 8) "500 ppm" replace with "500 ppb". It is reasonable to explain the term "Immediate Exposure" (page 13).

Author Response

We thanks all four reviewers for their time and helpful comments. We believe that we have addressed all the comments raised. We have rearranged the sections of the paper in a more logical fashion and hope that readers will now find the manuscript clearer and more concise. We have reproduced all the comments below in bold and given our response to each point in regular text below.

Reviewer 3

Comments and Suggestions for Authors

Abstract: "WASPSS" replace with "Western Air-Shed Particulate Study for Sidney".

We have removed the reference to WASPSS from the abstract and now it is described first in the introduction.

Misprint (page 8) "500 ppm" replace with "500 ppb".

Line 222 this has been corrected

It is reasonable to explain the term "Immediate Exposure" (page 13).

Suggest an introductory sentence on immediate vs cumulative exposure, with reference 

We have added the following sentences as an introduction to these terms: Exposure to poor air quality can have immediate negative health impacts, such as increased hospital admissions during pollution events (e.g.cite{johnston2011,dennekamp2015}).  However there is also evidence of increased morbidity as a result of cumulative exposure to air pollutants (e.g.\cite{Caldwell1998, reid2016, shah2013}). Therefore both immediate and cumulative exposure are important when considering the impacts of smoke on air quality in Sydney.

Reviewer 4 Report

Review of "atmosphere-576472" by Desservettaz et al. 

In this manuscript, the authors have analyzed the smoke events and their impact on air quality. Overall, the study is interesting and can be considered for publication in Atmosphere after revision. 

The abstract is confusing and not clear. It is recommended to revise the manuscript based on the following sequence: (i) define the problem, (iii) mention the data used, (iii) describe the methodology, and (iv) discuss the findings of this study.  It is also recommended to define all the abbreviation used in the abstract/manuscript.  Replace find at lines 9 and 10 with found.  it is better to avoid using the first pronoun in scientific writings.  Divide section 2 into parts: Data collection/used and methods. Sections 2.1 to 2.3 are related to data collections/used.  it is recommended to add gridlines on each map. 

Author Response

We thanks all four reviewers for their time and helpful comments. We believe that we have addressed all the comments raised. We have rearranged the sections of the paper in a more logical fashion and hope that readers will now find the manuscript clearer and more concise. We have reproduced all the comments below in bold and given our response to each point in regular text below.

Reviewer 4

Comments and Suggestions for Authors

Review of "atmosphere-576472" by Desservettaz et al. 

In this manuscript, the authors have analyzed the smoke events and their impact on air quality. Overall, the study is interesting and can be considered for publication in Atmosphere after revision. 

The abstract is confusing and not clear.

We have edited the Abstract and hope that it is clearer now.

Replace find at lines 9 and 10 with found.

We rephrased to remove the term here.

It is better to avoid using the first pronoun in scientific writings. 

This is a matter of personal style. We have removed some uses of the personal pronoun in the manuscript but left others in where the flow of the text benefitted from its use.

It is also recommended to define all the abbreviation used in the abstract/manuscript. 

We have revised the manuscript and think we have achieved this now.

It is recommended to add gridlines on each map.  

Again this is a matter of preference and we do not agree that this would add clarity.

It is recommended to revise the manuscript based on the following sequence: (i) define the problem, (iii) mention the data used, (iii) describe the methodology, and (iv) discuss the findings of this study. 

On first reading this comment seemed a little strange since the methodology includes making measurements in order to provide data for the study. Nevertheless, through trying to understand this comment we realised that the structure of the manuscript was disjointed and so we have undertaken a major reordering of the sections. We hope that the restructure of the manuscript (and change of the sub-headings) add clarity to the paper.

Divide section 2 into parts: Data collection/used and methods. Sections 2.1 to 2.3 are related to data collections/used. 

See comments about restructure of the manuscript above.

Round 2

Reviewer 1 Report

Reviewer comments

No: atmosphere-576472

Title: Air quality impacts of smoke from hazard reduction burns and domestic wood heating in western Sydney

This MS has been improved by reviewer comments. Therefore, the MS is recommend to publish in Atmosphere in present form.

Author Response

We thanks the reviewer for their time and effort in helping us to improve this manuscript.

Reviewer 4 Report

The manuscript can be accepted for publication but I would like to add comments on the authors' responses.

Regarding the first pronoun: I would recommend the authors do some research and learn how to write scientific articles - it will be good for them for their next publication.  The authors were asked to add grid-lines but they respond that it is a personal matter. I would suggest the authors learn the basic of the map, it would be useful for them and encourage them to add grid-lies.  Again, my suggestion is to learn research methods and find the differences between data collection and research methodology. 

Author Response

We thank the reviewer for their time and efforts in helping us to improve this manuscript. Since this reviewer now states that the manuscript can be accepted for publication, we hope that the editor will choose to do so.

The reviewer has provided some advice for future manuscripts, however the authors are not in full agreement with the views expressed by this reviewer on how best to present a scientific manuscript. The remaining points of issue are all stylistic and in my opinion the style should be chosen for clarity – and this does not exclude use of the first pronoun or maps without grid-lines.

I have to admit to being somewhat surprised by these comments and so I did a quick search in Web of Science using the term “Atmospheric Chemistry”, and chose the most highly cited papers. I then choose the first paper in Science and the first in Atmospheric Chemistry and Physics from the top five most cited papers. Both these papers use the first pronoun in the Abstract, confirming my impression that this is not unusual in scientific papers in the field. Thus I assume that the reviewer’s reading is in a somewhat different area to my own. (These two papers are referenced below).

Jimenez, J.L.; Canagaratna, M.R.; Donahue, N.M.; Prevot, A.S.H.; Zhang, Q.; Kroll, J.H.; DeCarlo, P.F.; Allan, J.D.; Coe, H.; Ng, N.L., et al. Evolution of Organic Aerosols in the Atmosphere. Science 2009, 326, 1525-1529, doi:10.1126/science.1180353. Lamarque, J.F.; Bond, T.C.; Eyring, V.; Granier, C.; Heil, A.; Klimont, Z.; Lee, D.; Liousse, C.; Mieville, A.; Owen, B., et al. Historical (1850-2000) gridded anthropogenic and biomass burning emissions of reactive gases and aerosols: methodology and application. Atmospheric Chemistry and Physics 2010, 10, 7017-7039, doi:10.5194/acp-10-7017-2010.

In short, the reviewer has stated that the paper may be accepted and we are not minded to make further changes to the manuscript because we are not convinced that they will improve the clarity of the paper.

Nevertheless, we appreciate the time and effort that went into providing this review and we acknowledge that the paper is now better for the original advice of this reviewer.